# Sequential Monte Carlo for Graphical Models

**Christian A. Naesseth**
Div. of Automatic Control
Linköping University
Linköping, Sweden
chran60@isy.liu.se

**Fredrik Lindsten**
Dept. of Engineering
The University of Cambridge
Cambridge, UK
fsml2@cam.ac.uk

**Thomas B. Schön**
Dept. of Information Technology
Uppsala University
Uppsala, Sweden
thomas.schon@it.uu.se

## Abstract

We propose a new framework for how to use sequential Monte Carlo (SMC) algorithms for inference in probabilistic graphical models (PGM). Via a sequential decomposition of the PGM we find a sequence of auxiliary distributions defined on a monotonically increasing sequence of probability spaces. By targeting these auxiliary distributions using SMC we are able to approximate the full joint distribution defined by the PGM. One of the key merits of the SMC sampler is that it provides an unbiased estimate of the partition function of the model. We also show how it can be used within a particle Markov chain Monte Carlo framework in order to construct high-dimensional block-sampling algorithms for general PGMs.

## 1 Introduction

Bayesian inference in statistical models involving a large number of latent random variables is in general a difficult problem. This renders inference methods that are capable of efficiently utilizing structure important tools. Probabilistic Graphical Models (PGMs) are an intuitive and useful way to represent and make use of underlying structure in probability distributions with many interesting areas of applications [1].

Our main contribution is a new framework for constructing non-standard (auxiliary) target distributions of PGMs, utilizing what we call a *sequential decomposition* of the underlying factor graph, to be targeted by a sequential Monte Carlo (SMC) sampler. This construction enables us to make use of SMC methods developed and studied over the last 20 years, to approximate the full joint distribution defined by the PGM. As a byproduct, the SMC algorithm provides an unbiased estimate of the partition function (normalization constant). We show how the proposed method can be used as an alternative to standard methods such as the Annealed Importance Sampling (AIS) proposed in [2], when estimating the partition function. We also make use of the proposed SMC algorithm to design efficient, high-dimensional MCMC kernels for the latent variables of the PGM in a particle MCMC framework. This enables inference about the latent variables as well as learning of unknown model parameters in an MCMC setting.

During the last decade there has been substantial work on how to leverage SMC algorithms [3] to solve inference problems in PGMs. The first approaches were PAMPAS [4] and nonparametric belief propagation by Sudderth et al. [5, 6]. Since then, several different variants and refinements have been proposed by e.g. Briers et al. [7], Ihler and Mcallester [8], Frank et al. [9]. They all rely on various particle approximations of messages sent in a loopy belief propagation algorithm. This means that in general, even in the limit of Monte Carlo samples, they are approximate methods. Compared to these approaches our proposed methods are consistent and provide an unbiased estimate of the normalization constant as a by-product.

Another branch of SMC-based methods for graphical models has been suggested by Hamze and de Freitas [10]. Their method builds on the SMC sampler by Del Moral et al. [11], where the

initial target is a spanning tree of the original graph and subsequent steps add edges according to an annealing schedule. Everitt [12] extends these ideas to learn parameters using particle MCMC [13]. Yet another take is provided by Carbonetto and de Freitas [14], where an SMC sampler is combined with mean field approximations. Compared to these methods we can handle both non-Gaussian and/or non-discrete interactions between variables and there is no requirement to perform MCMC steps within each SMC step.

The left-right methods described by Wallach et al. [15] and extended by Buntine [16] to estimate the likelihood of held-out documents in topic models are somewhat related in that they are SMC-inspired. However, these are not actual SMC algorithms and they do not produce an unbiased estimate of the partition function for finite sample set. On the other hand, a particle learning based approach was recently proposed by Scott and Baldridge [17] and it can be viewed as a special case of our method for this specific type of model.

## 2  Graphical models

A graphical model is a probabilistic model which *factorizes* according to the structure of an underlying graph $\mathcal{G} = \{\mathcal{V}, \mathcal{E}\}$, with vertex set $\mathcal{V}$ and edge set $\mathcal{E}$. By this we mean that the joint probability density function (PDF) of the set of random variables indexed by $\mathcal{V}$, $X_{\mathcal{V}} := \{x_1, \ldots, x_{|\mathcal{V}|}\}$, can be represented as a product of factors over the cliques of the graph:

$$p(X_{\mathcal{V}}) = \frac{1}{Z} \prod_{C \in \mathcal{C}} \psi_C(X_C), \tag{1}$$

where $\mathcal{C}$ is the set of cliques in $\mathcal{G}$, $\psi_C$ is the factor for clique $C$, and $Z = \int \prod_{C \in \mathcal{C}} \psi_C(x_C) \mathrm{d}X_{\mathcal{V}}$ is the partition function.

We will frequently use the notation $X_I = \bigcup_{i \in I} \{x_i\}$ for some subset $I \subseteq \{1, \ldots, |\mathcal{V}|\}$ and we write $\mathsf{X}_I$ for the range of $X_I$ (i.e., $X_I \in \mathsf{X}_I$). To make the interactions between the random variables explicit we define a *factor graph* $\mathcal{F} = \{\mathcal{V}, \Psi, \mathcal{E}'\}$ corresponding to $\mathcal{G}$. The factor graph consists of two types of vertices, the original set of random variables $X_{\mathcal{V}}$ and the factors $\Psi = \{\psi_C : C \in \mathcal{C}\}$. The edge set $\mathcal{E}'$ consists only of edges from variables to factors. In Figure 1a we show a simple toy example of an undirected graphical model, and one possible corresponding factor graph, Figure 1b, making the dependencies explicit. Both directed and undirected graphs can be represented by factor graphs.

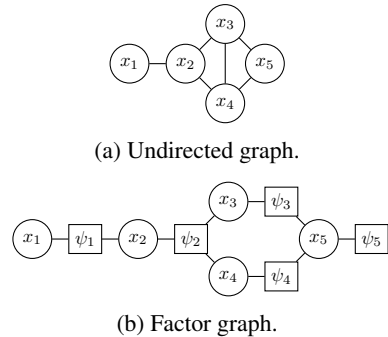

(a) Undirected graph.

(b) Factor graph.

Figure 1: Undirected PGM and a corresponding factor graph.

## 3  Sequential Monte Carlo

In this section we propose a way to sequentially decompose a graphical model which we then make use of to design an SMC algorithm for the PGM.

### 3.1  Sequential decomposition of graphical models

SMC methods can be used to approximate a sequence of probability distributions on a sequence of probability spaces of increasing dimension. This is done by recursively updating a set of samples—or *particles*—with corresponding nonnegative importance weights. The typical scenario is that of state inference in state-space models, where the probability distributions targeted by the SMC sampler are the joint smoothing distributions of a sequence of latent states conditionally on a sequence of observations; see e.g., Doucet and Johansen [18] for applications of this type. However, SMC is not limited to these cases and it is applicable to a much wider class of models.

To be able to use SMC for inference in PGMs we have to define a sequence of target distributions. However, these target distributions *do not* have to be marginal distributions under $p(X_{\mathcal{V}})$. Indeed, as long as the sequence of target distributions is constructed in such a way that, at some final iteration, we recover $p(X_{\mathcal{V}})$, all the intermediate target distributions may be chosen quite arbitrarily.

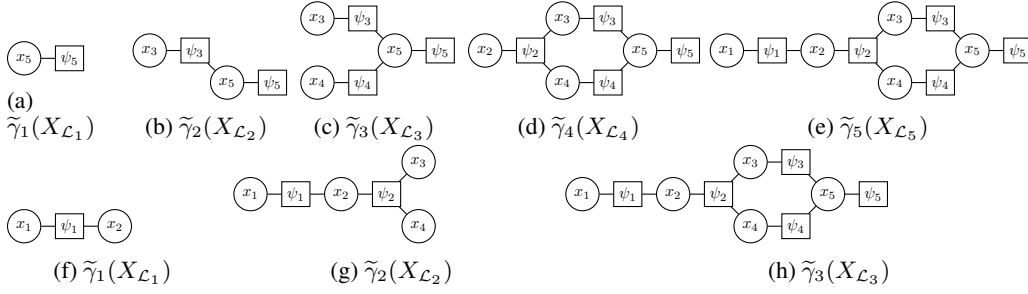

Figure 2: Examples of five- (top) and three-step (bottom) sequential decomposition of Figure 1.

This is key to our development, since it lets us use the structure of the PGM to define a sequence of intermediate target distributions for the sampler. We do this by a so called *sequential decomposition* of the graphical model. This amounts to simply adding factors to the target distribution, from the product of factors in (1), at each step of the algorithm and iterate until all the factors have been added. Constructing an artificial sequence of intermediate target distributions for an SMC sampler is a simple, albeit underutilized, idea as it opens up for using SMC samplers for inference in a wide range of probabilistic models; see e.g., Bouchard-Côté et al. [19], Del Moral et al. [11] for a few applications of this approach.

Given a graph $\mathcal{G}$ with cliques $\mathcal{C}$, let $\{\psi_k\}_{k=1}^K$ be a sequence of factors defined as follows $\psi_k(X_{\mathcal{I}_k}) = \prod_{C \in \mathcal{C}_k} \psi_C(X_C)$, where $\mathcal{C}_k \subset \mathcal{C}$ are chosen such that $\bigcup_{k=1}^K \mathcal{C}_k = \mathcal{C}$ and $\mathcal{C}_i \cap \mathcal{C}_j = \emptyset$, $i \neq j$, and where $\mathcal{I}_k \subseteq \{1, \ldots, |\mathcal{V}|\}$ is the index set of the variables in the domain of $\psi_k$, $\mathcal{I}_k = \bigcup_{C \in \mathcal{C}_k} C$. We emphasize that the cliques in $\mathcal{C}$ need not be maximal. In fact even auxiliary factors may be introduced to allow for e.g. annealing between distributions. It follows that the PDF in (1) can be written as $p(X_{\mathcal{V}}) = \frac{1}{Z} \prod_{k=1}^K \psi_k(X_{\mathcal{I}_k})$. Principally, the choices and the ordering of the $\mathcal{C}_k$'s is arbitrary, but in practice it will affect the performance of the proposed sampler. However, in many common PGMs an intuitive ordering can be deduced from the structure of the model, see Section 5.

The sequential decomposition of the PGM is then based on the auxiliary quantities $\widetilde{\gamma}_k(X_{\mathcal{L}_k}) := \prod_{\ell=1}^k \psi_\ell(X_{\mathcal{I}_\ell})$, with $\mathcal{L}_k := \bigcup_{\ell=1}^k \mathcal{I}_\ell$, for $k \in \{1, \ldots, K\}$. By construction, $\mathcal{L}_K = \mathcal{V}$ and the joint PDF $p(X_{\mathcal{L}_K})$ will be proportional to $\widetilde{\gamma}_K(X_{\mathcal{L}_K})$. Consequently, by using $\widetilde{\gamma}_k(X_{\mathcal{L}_k})$ as the basis for the target sequence for an SMC sampler, we will obtain the correct target distribution at iteration $K$. However, a further requirement for this to be possible is that all the functions in the sequence are normalizable. For many graphical models this is indeed the case, and then we can use $\widetilde{\gamma}_k(X_{\mathcal{L}_k})$, $k = 1$ to $K$, directly as our sequence of intermediate target densities. If, however, $\int \widetilde{\gamma}_k(X_{\mathcal{L}_k}) \mathrm{d}X_{\mathcal{L}_k} = \infty$ for some $k < K$, an easy remedy is to modify the target density to ensure normalizability. This is done by setting $\gamma_k(X_{\mathcal{L}_k}) = \widetilde{\gamma}_k(X_{\mathcal{L}_k}) q_k(X_{\mathcal{L}_k})$, where $q_k(X_{\mathcal{L}_k})$ is choosen so that $\int \gamma_k(X_{\mathcal{L}_k}) \mathrm{d}X_{\mathcal{L}_k} < \infty$. We set $q_K(X_{\mathcal{L}_K}) \equiv 1$ to make sure that $\gamma_K(X_{\mathcal{L}_K}) \propto p(X_{\mathcal{L}_K})$. Note that the integral $\int \gamma_k(X_{\mathcal{L}_k}) \mathrm{d}X_{\mathcal{L}_k}$ need not be computed explicitly, as long as it can be established that it is finite. With this modification we obtain a sequence of unnormalized intermediate target densities for the SMC sampler as $\gamma_1(X_{\mathcal{L}_1}) = q_1(X_{\mathcal{L}_1}) \psi_1(X_{\mathcal{L}_1})$ and $\gamma_k(X_{\mathcal{L}_k}) = \gamma_{k-1}(X_{\mathcal{L}_{k-1}}) \frac{q_k(X_{\mathcal{L}_k})}{q_{k-1}(X_{\mathcal{L}_{k-1}})} \psi_k(X_{\mathcal{I}_k})$ for $k = 2, \ldots, K$. The corresponding normalized PDFs are given by $\bar{\gamma}_k(X_{\mathcal{L}_k}) = \gamma_k(X_{\mathcal{L}_k})/Z_k$, where $Z_k = \int \gamma_k(X_{\mathcal{L}_k}) \mathrm{d}X_{\mathcal{L}_k}$. Figure 2 shows two examples of possible subgraphs when applying the decomposition, in two different ways, to the factor graph example in Figure 1.

## 3.2 Sequential Monte Carlo for PGMs

At iteration $k$, the SMC sampler approximates the target distribution $\bar{\gamma}_k$ by a collection of weighted particles $\{X_{\mathcal{L}_k}^i, w_k^i\}_{i=1}^N$. These samples define an empirical point-mass approximation of the target distribution. In what follows, we shall use the notation $\xi_k := X_{\mathcal{I}_k \backslash \mathcal{L}_{k-1}}$ to refer to the collection of random variables that are in the domain of $\gamma_k$, but not in the domain of $\gamma_{k-1}$. This corresponds to the collection of random variables, with which the particles are augmented at each iteration.

Initially, $\bar{\gamma}_1$ is approximated by importance sampling. We proceed inductively and assume that we have at hand a weighted sample $\{X_{\mathcal{L}_{k-1}}^i, w_{k-1}^i\}_{i=1}^N$, approximating $\bar{\gamma}_{k-1}(X_{\mathcal{L}_{k-1}})$. This sample is

propagated forward by simulating, conditionally independently given the particle generation up to iteration $k-1$, and drawing an *ancestor index* $a_k^i$ with $\mathbb{P}(a_k^i = j) \propto \nu_{k-1}^j w_{k-1}^j$, $j = 1, \ldots, N$, where $\nu_{k-1}^i := \nu_{k-1}(X_{\mathcal{L}_{k-1}}^i)$—known as adjustment multiplier weights—are used in the auxiliary SMC framework to adapt the resampling procedure to the current target density $\bar{\gamma}_k$ [20]. Given the ancestor indices, we simulate particle increments $\{\xi_k^i\}_{i=1}^N$ from a proposal density $\xi_k^i \sim r_k(\cdot | X_{\mathcal{L}_{k-1}}^{a_k^i})$ on $\mathsf{X}_{\mathcal{I}_k \setminus \mathcal{L}_{k-1}}$, and augment the particles as $X_{\mathcal{L}_k}^i := X_{\mathcal{L}_{k-1}}^{a_k^i} \cup \xi_k^i$.

After having performed this procedure for the $N$ ancestor indices and particles, they are assigned importance weights $w_k^i = W_k(X_{\mathcal{L}_k}^i)$. The weight function, for $k \geq 2$, is given by

$$W_k(X_{\mathcal{L}_k}) = \frac{\gamma_k(X_{\mathcal{L}_k})}{\gamma_{k-1}(X_{\mathcal{L}_{k-1}})\nu_{k-1}(X_{\mathcal{L}_{k-1}})r_k(\xi_k | X_{\mathcal{L}_{k-1}})}, \tag{2}$$

where, again, we write $\xi_k = X_{\mathcal{I}_k \setminus \mathcal{L}_{k-1}}$. We give a summary of the SMC method in Algorithm 1.

In the case that $\mathcal{I}_k \setminus \mathcal{L}_{k-1} = \emptyset$ for some $k$, resampling and propagation steps are superfluous. The easiest way to handle this is to simply skip these steps and directly compute importance weights. An alternative approach is to bridge the two target distributions $\bar{\gamma}_{k-1}$ and $\bar{\gamma}_k$ similarly to Del Moral et al. [11].

Since the proposed sampler for PGMs falls within a general SMC framework, standard convergence

---

**Algorithm 1** Sequential Monte Carlo (SMC)

*Perform each step for $i = 1, \ldots, N$.*
Sample $X_{\mathcal{L}_1}^i \sim r_1(\cdot)$.
Set $w_1^i = \gamma_1(X_{\mathcal{L}_1}^i)/r_1(X_{\mathcal{L}_1}^i)$.
**for** $k = 2$ **to** $K$ **do**
    Sample $a_k^i$ according to $\mathbb{P}(a_k^i = j) = \frac{\nu_{k-1}^j w_{k-1}^j}{\sum_l \nu_{k-1}^l w_{k-1}^l}$.
    Sample $\xi_k^i \sim r_k(\cdot | X_{\mathcal{L}_{k-1}}^{a_k^i})$ and set $X_{\mathcal{L}_k}^i = X_{\mathcal{L}_{k-1}}^{a_k^i} \cup \xi_k^i$.
    Set $w_k^i = W_k(X_{\mathcal{L}_k}^i)$.
**end for**

---

analysis applies. See e.g., Del Moral [21] for a comprehensive collection of theoretical results on consistency, central limit theorems, and non-asymptotic bounds for SMC samplers.

The choices of proposal density and adjustment multipliers can quite significantly affect the performance of the sampler. It follows from (2) that $W_k(X_{\mathcal{L}_k}) \equiv 1$ if we choose $\nu_{k-1}(X_{\mathcal{L}_{k-1}}) = \int \frac{\gamma_k(X_{\mathcal{L}_k})}{\gamma_{k-1}(X_{\mathcal{L}_{k-1}})} \mathrm{d}\xi_k$ and $r_k(\xi_k | X_{\mathcal{L}_{k-1}}) = \frac{\gamma_k(X_{\mathcal{L}_k})}{\nu_{k-1}(X_{\mathcal{L}_{k-1}})\gamma_{k-1}(X_{\mathcal{L}_{k-1}})}$. In this case, the SMC sampler is said to be *fully adapted*.

### 3.3 Estimating the partition function

The partition function of a graphical model is a very interesting quantity in many applications. Examples include likelihood-based learning of the parameters of the PGM, statistical mechanics where it is related to the free energy of a system of objects, and information theory where it is related to the capacity of a channel. However, as stated by Hamze and de Freitas [10], estimating the partition function of a loopy graphical model is a "notoriously difficult" task. Indeed, even for discrete problems simple and accurate estimators have proved to be elusive, and MCMC methods do not provide any simple way of computing the partition function.

On the contrary, SMC provides a straightforward estimator of the normalizing constant (i.e. the partition function), given as a byproduct of the sampler according to,

$$\widehat{Z}_k^N := \left( \frac{1}{N} \sum_{i=1}^N w_k^i \right) \left\{ \prod_{\ell=1}^{k-1} \frac{1}{N} \sum_{i=1}^N \nu_\ell^i w_\ell^i \right\}. \tag{3}$$

It may not be obvious to see why (3) is a natural estimator of the normalizing constant $Z_k$. However, a by now well known result is that this SMC-based estimator is unbiased. This result is due to Del Moral [21, Proposition 7.4.1] and, for the special case of inference in state-space models, it has also been established by Pitt et al. [22]. For completeness we also offer a proof using the present notation in the supplementary material. Since $Z_K = Z$, we thus obtain an estimator of the partition function of the PGM at iteration $K$ of the sampler. Besides from being unbiased, this estimator is also consistent and asymptotically normal; see Del Moral [21].

In [23] we have studied a specific information theoretic application (computing the capacity of a two-dimensional channel) and inspired by the algorithm proposed here we were able to design a sampler with significantly improved performance compared to the previous state-of-the-art.

# 4 Particle MCMC and partial blocking

Two shortcomings of SMC are: *(i)* it does not solve the parameter learning problem, and *(ii)* the quality of the estimates of marginal distributions $p(X_{\mathcal{L}_k}) = \int \bar{\gamma}_K(X_{\mathcal{L}_K}) \mathrm{d}X_{\mathcal{L}_K \backslash \mathcal{L}_k}$ deteriorates for $k \ll K$ due to the fact that the particle trajectories degenerate as the particle system evolves (see e.g., [18]). Many methods have been proposed in the literature to address these problems; see e.g. [24] and the references therein. Among these, the recently proposed particle MCMC (PMCMC) framework [13], plays a prominent role. PMCMC algorithms make use of SMC to construct (in general) high-dimensional Markov kernels that can be used within MCMC. These methods were shown by [13] to be exact, in the sense that the apparent particle approximation in the construction of the kernel does not change its invariant distribution. This property holds for any number of particles $N \geq 2$, i.e., PMCMC does not rely on asymptotics in $N$ for correctness.

The fact that the SMC sampler for PGMs presented in Algorithm 1 fits under a general SMC umbrella implies that we can also straightforwardly make use of this algorithm within PMCMC. This allows us to construct a Markov kernel (indexed by the number of particles $N$) on the space of latent variables of the PGM, $P_N(X'_{\mathcal{L}_K}, \mathrm{d}X_{\mathcal{L}_K})$, which leaves the full joint distribution $p(X_{\mathcal{V}})$ invariant. We do not dwell on the details of the implementation here, but refer instead to [13] for the general setup and [25] for the specific method that we have used in the numerical illustration in Section 5.

PMCMC methods enable blocking of the latent variables of the PGM in an MCMC scheme. Simulating all the latent variables $X_{\mathcal{L}_K}$ jointly is useful since, in general, this will reduce the autocorrelation when compared to simulating the variables $x_j$ one at a time [26]. However, it is also possible to employ PMCMC to construct an algorithm in between these two extremes, a strategy that we believe will be particularly useful in the context of PGMs. Let $\{\mathcal{V}^m, \, m \in \{1, \ldots, M\}\}$ be a partition of $\mathcal{V}$. Ideally, a Gibbs sampler for the joint distribution $p(X_{\mathcal{V}})$ could then be constructed by simulating, using a systematic or a random scan, from the conditional distributions

$$p(X_{\mathcal{V}^m}|X_{\mathcal{V} \backslash \mathcal{V}^m}) \text{ for } m = 1, \ldots, M. \tag{4}$$

We refer to this strategy as *partial blocking*, since it amounts to simulating a subset of the variables, but not necessarily all of them, jointly. Note that, if we set $M = |\mathcal{V}|$ and $\mathcal{V}^m = \{m\}$ for $m = 1, \ldots, M$, this scheme reduces to a standard Gibbs sampler. On the other extreme, with $M = 1$ and $\mathcal{V}^1 = \mathcal{V}$, we get a fully blocked sampler which targets directly the full joint distribution $p(X_{\mathcal{V}})$.

From (1) it follows that the conditional distributions (4) can be expressed as

$$p(X_{\mathcal{V}^m}|X_{\mathcal{V} \backslash \mathcal{V}^m}) \propto \prod_{C \in \mathcal{C}^m} \psi_C(X_C), \tag{5}$$

where $\mathcal{C}^m = \{C \in \mathcal{C} : C \cap \mathcal{V}^m \neq \emptyset\}$. While it is in general not possible to sample exactly from these conditionals, we can make use of PMCMC to facilitate a partially blocked Gibbs sampler for a PGM. By letting $p(X_{\mathcal{V}^m}|X_{\mathcal{V} \backslash \mathcal{V}^m})$ be the target distribution for the SMC sampler of Algorithm 1, we can construct a PMCMC kernel $P_N^m$ that leaves the conditional distribution (5) invariant. This suggests the following approach: with $X'_{\mathcal{V}}$ being the current state of the Markov chain, update block $m$ by sampling

$$X_{\mathcal{V}^m} \sim P_N^m \langle X'_{\mathcal{V} \backslash \mathcal{V}^m} \rangle (X'_{\mathcal{V}^m}, \cdot). \tag{6}$$

Here we have indicated explicitly in the notation that the PMCMC kernel for the conditional distribution $p(X_{\mathcal{V}^m}|X_{\mathcal{V} \backslash \mathcal{V}^m})$ depends on both $X'_{\mathcal{V} \backslash \mathcal{V}^m}$ (which is considered to be fixed throughout the sampling procedure) and on $X'_{\mathcal{V}^m}$ (which defines the current state of the PMCMC procedure).

As mentioned above, while being generally applicable, we believe that partial blocking of PMCMC samplers will be particularly useful for PGMs. The reason is that we can choose the vertex sets $\mathcal{V}^m$ for $m = 1, \ldots, M$ in order to facilitate simple sequential decompositions of the induced subgraphs. For instance, it is always possible to choose the partition in such a way that all the induced subgraphs are chains.

# 5  Experiments

In this section we evaluate the proposed SMC sampler on three examples to illustrate the merits of our approach. Additional details and results are available in the supplementary material and code to reproduce results can be found in [27]. We first consider an example from statistical mechanics, the classical XY model, to illustrate the impact of the sequential decomposition. Furthermore, we profile our algorithm with the "gold standard" AIS [2] and Annealed Sequential Importance Resampling (ASIR[1]) [11]. In the second example we apply the proposed method to the problem of scoring of topic models, and finally we consider a simple toy model, a Gaussian Markov random field (MRF), which illustrates that our proposed method has the potential to significantly decrease correlations between samples in an MCMC scheme. Furthermore, we provide an *exact* SMC-approximation of the tree-sampler by Hamze and de Freitas [28] and thereby extend the scope of this powerful method.

## 5.1  Classical XY model

The classical XY model (see e.g. [29]) is a member in the family of *n-vector* models used in statistical mechanics. It can be seen as a generalization of the well known Ising model with a two-dimensional electromagnetic spin. The spin vector is described by its angle $x \in (-\pi, \pi]$. We will consider square lattices with periodic boundary conditions. The joint PDF of the classical XY model with equal interaction is given by

$$p(X_\mathcal{V}) \propto e^{\beta \sum_{(i,j) \in \mathcal{E}} \cos(x_i - x_j)}, \qquad (7)$$

where $\beta$ denotes the inverse temperature.

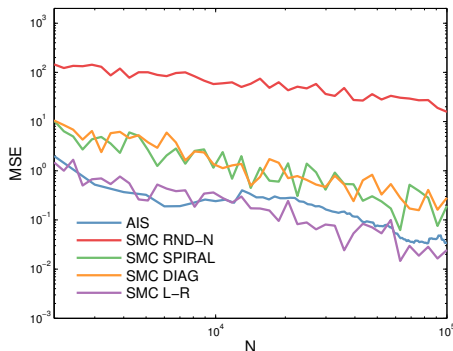

Figure 3: Mean-squared-errors for sample size $N$ in the estimates of $\log Z$ for AIS and four different orderings in the proposed SMC framework.

To evaluate the effect of different sequence orders on the accuracy of the estimates of the log-normalizing-constant $\log Z$ we ran several experiments on a $16 \times 16$ XY model with $\beta = 1.1$ (approximately the critical inverse temperature [30]). For simplicity we add one node at a time and all factors bridging this node with previously added nodes. Full adaptation in this case is possible due to the optimal proposal being a von Mises distribution. We show results for the following cases: *Random neighbour (RND-N)* First node selected randomly among all nodes, concurrent nodes selected randomly from the set of nodes with a neighbour in $X_{\mathcal{L}_{k-1}}$. *Diagonal (DIAG)* Nodes added by traversing diagonally (45° angle) from left to right. *Spiral (SPIRAL)* Nodes added spiralling in towards the middle from the edges. *Left-Right (L-R)* Nodes added by traversing the graph left to right, from top to bottom.

We also give results of AIS with single-site-Gibbs updates and $1\,000$ annealing distributions linearly spaced from zero to one, starting from a uniform distribution (geometric spacing did not yield any improvement over linear spacing for this case). The "true value" was estimated using AIS with $10\,000$ intermediate distributions and $5\,000$ importance samples. We can see from the results in Figure 3 that designing a good sequential decomposition for the SMC sampler is important. However, the intuitive and fairly simple choice L-R does give very good results comparable to that of AIS.

Furthermore, we consider a larger size of $64 \times 64$ and evaluate the performance of the L-R ordering compared to AIS and the ASIR method. Figure 4 displays box-plots of 10 independent runs. We set $N = 10^5$ for the proposed SMC sampler and then match the computational costs of AIS and ASIR with this computational budget. A fair amount of time was spent in tuning the AIS and ASIR algorithms; $10\,000$ linear annealing distributions seemed to give best performance in these cases. We can see that the L-R ordering gives results comparable to fairly well-tuned AIS and ASIR algorithms; the ordering of the methods depending on the temperature of the model. One option that does make the SMC algorithm interesting for these types of applications is that it can easily be parallelized over the particles, whereas AIS/ASIR has limited possibilities of parallel implementation over the (crucial) annealing steps.

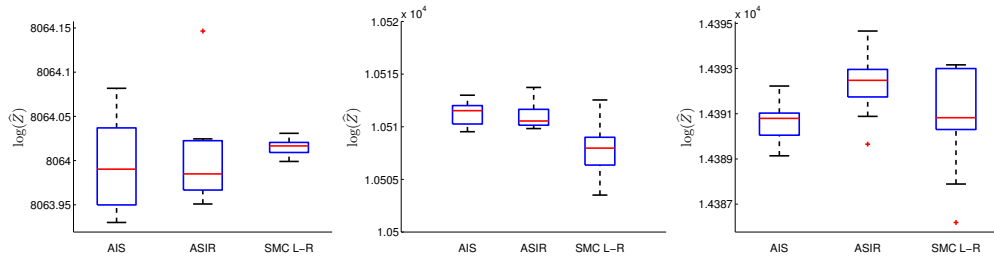

Figure 4: The logarithm of the estimated partition function for the $64 \times 64$ XY model with inverse temperature $0.5$ (left), $1.1$ (middle) and $1.7$ (right).

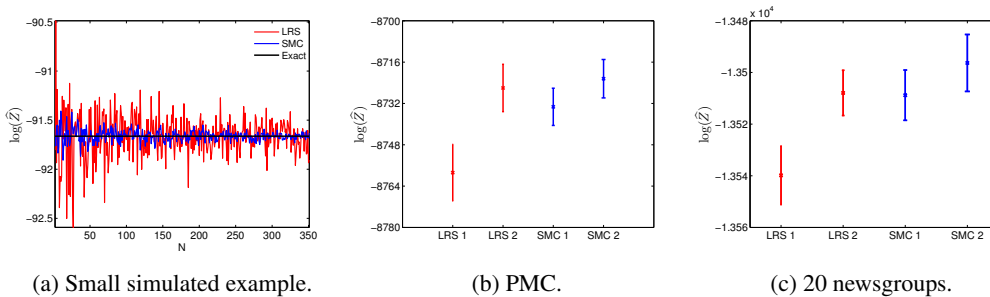

(a) Small simulated example.   (b) PMC.   (c) 20 newsgroups.

Figure 6: Estimates of the log-likelihood of heldout documents for various datasets.

## 5.2   Likelihood estimation in topic models

Topic models such as Latent Dirichlet Allocation (LDA) [31] are popular models for reasoning about large text corpora. Model evaluation is often conducted by computing the likelihood of held-out documents w.r.t. a learnt model. However, this is a challenging problem on its own—which has received much recent interest [15, 16, 17]—since it essentially corresponds to computing the partition function of a graphical model; see Figure 5. The SMC procedure of Algorithm 1 can used to solve this problem by defining a sequential decomposition of the graphical model. In particular, we consider the decomposition corresponding to first including the node $\theta$ and then, subsequently, introducing the nodes $z_1$ to $z_M$ in any order. Interestingly, if we then make use of a Rao-Blackwellization over the variable $\theta$, the SMC sampler of Algorithm 1 reduces exactly to a method that has previously been proposed for this specific problem [17]. In [17], the method is derived by reformulating the model in terms of its sufficient statistics and phrasing this as a particle learning problem; here we obtain the same procedure as a special case of the general SMC algorithm operating on the original model.

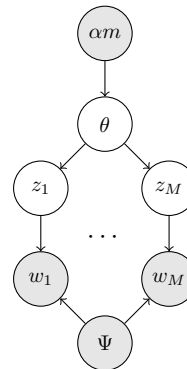

Figure 5: LDA as graphical model.

We use the same data and learnt models as Wallach et al. [15], i.e. 20 newsgroups, and PubMed Central abstracts (PMC). We compare with the Left-Right-Sequential (LRS) sampler [16], which is an improvement over the method proposed by Wallach et al. [15]. Results on simulated and real data experiments are provided in Figure 6. For the simulated example (Figure 6a), we use a small model with 10 words and 4 topics to be able to compute the exact log-likelihood. We keep the number of particles in the SMC algorithm equal to the number of Gibbs steps in LRS; this means LRS is about an order-of-magnitude more computationally demanding than the SMC method. Despite the fact that the SMC sampler uses only about a tenth of the computational time of the LRS sampler, it performs significantly better in terms of estimator variance. The other two plots show results on real data with 10 held-out documents for each dataset. For a fixed number of Gibbs steps we choose the number of particles for each document to make the computational cost approximately equal. Run #2 has twice the number of particles/samples as in run #1. We show the mean of 10 runs and error-bars estimated

using bootstrapping with $10\,000$ samples. Computing the logarithm of $\hat{Z}$ introduces a negative bias, which means larger values of $\log \hat{Z}$ typically implies more accurate results. The results on real data do not show the drastic improvement we see in the simulated example, which could be due to degeneracy problems for long documents. An interesting approach that could improve results would be to use an SMC algorithm tailored to discrete distributions, e.g. Fearnhead and Clifford [32].

### 5.3 Gaussian MRF

Finally, we consider a simple toy model to illustrate how the SMC sampler of Algorithm 1 can be incorporated in PMCMC sampling. We simulate data from a zero mean Gaussian $10 \times 10$ lattice MRF with observation and interaction standard deviations of $\sigma_i = 1$ and $\sigma_{ij} = 0.1$ respectively. We use the proposed SMC algorithm together with the PMCMC method by Lindsten et al. [25]. We compare this with standard Gibbs sampling and the tree sampler by Hamze and de Freitas [28].

We use a moderate number of $N = 50$ particles in the PMCMC sampler (recall that it admits the correct invariant distribution for any $N \geq 2$). In Figure 7 we can see the empirical autocorrelation funtions (ACF) centered around the true posterior mean for variable $x_{82}$ (selected randomly from among $X_{\mathcal{V}}$; similar results hold for all the variables of the model). Due to the strong interaction between the latent variables, the samples generated by the standard Gibbs sampler are strongly correlated. Tree-sampling and PMCMC with partial blocking show nearly identical gains compared to Gibbs. This is interesting, since it suggest that simulating from the SMC-based PMCMC kernel can be almost as efficient as exact simulation, even using a moderate number of particles. Indeed, PMCMC with partial blocking can be viewed as an *exact* SMC-approximation of the tree sampler, extending the scope of tree-sampling beyond discrete and Gaussian models. The fully blocked PMCMC al-

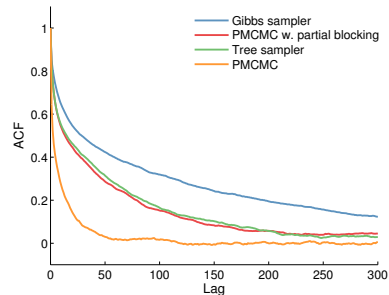

Figure 7: The empirical ACF for Gibbs sampling, PMCMC, PMCMC with partial blocking, and tree sampling.

gorithm achieves the best ACF, dropping off to zero considerably faster than for the other methods. This is not surprising since this sampler simulates all the latent variables jointly which reduces the autocorrelation, in particular when the latent variables are strongly dependent. However, it should be noted that this method also has the highest computational cost per iteration.

## 6 Conclusion

We have proposed a new framework for inference in PGMs using SMC and illustrated it on three examples. These examples show that it can be a viable alternative to standard methods used for inference and partition function estimation problems. An interesting avenue for future work is combining our proposed methods with AIS, to see if we can improve on both.

#### Acknowledgments

We would like to thank Iain Murray for his kind and very prompt help in providing the data for the LDA example. This work was supported by the projects: *Learning of complex dynamical systems* (Contract number: 637-2014-466) and *Probabilistic modeling of dynamical systems* (Contract number: 621-2013-5524), both funded by the Swedish Research Council.

## Footnotes

[1]ASIR is a specific instance of the *SMC sampler* by [11], corresponding to AIS with the addition of resampling steps, but to avoid confusion with the proposed method we choose to refer to it as ASIR.

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
