[Supplementary Material]

# Sequential Monte Carlo for Graphical Models – Supplementary Material

**Christian A. Naesseth**
Div. of Automatic Control
Linköping University
Linköping, Sweden
chran60@isy.liu.se

**Fredrik Lindsten**
Dept. of Engineering
The University of Cambridge
Cambridge, UK
fsml2@cam.ac.uk

**Thomas B. Schön**
Dept. of Information Technology
Uppsala University
Uppsala, Sweden
thomas.schon@it.uu.se

## Abstract

This supplementary material contains additional information on the experiments in the main paper [Naesseth et al., 2014] as well as a simple and direct proof of the unbiasedness of the partition function estimator $\widehat{Z}_k^N$, stated in the main manuscript. It should be noted, however, that this result is not new. It has previously been established in a general setting by Del Moral [2004, Proposition 7.4.1] and, additionally, by Pitt et al. [2012] who provide a more accessible proof for the special case of state-space models. Our proof is similar to that of Pitt et al. [2012], but generalized to the PGM setting that we consider.

## 1   Experiments

### 1.1   Classical XY model

The classical XY model, see e.g. [Kosterlitz and Thouless, 1973] and references therein, is a member in the family of *n-vector* models used in statistical mechanics. It can be seen as a generalization of the well known Ising model with a two-dimensional electromagnetic spin. The spin vector is described by its angle $x \in (-\pi, \pi]$. We will consider a square lattice with periodic boundary conditions, i.e. the first and last row/columns are connected. The individual sites are described by their spin angle.

The full joint PDF of the classical XY model is given by

$$p(X_\mathcal{V}) \propto e^{-\beta H(X_\mathcal{V})}, \tag{1}$$

where $\beta$ is the inverse temperature and $H(X_\mathcal{V})$—the Hamiltonian—is a sum of pair-wise interaction described by

$$H(X_\mathcal{V}) = -\sum_{(i,j)\in\mathcal{E}} J_{ij} \cos(x_i - x_j), \tag{2}$$

where the $J_{ij}$'s are parameters describing interactions between the different sites. For simplicity we set $J_{ij} = J = 1$ and estimate the partition function for several sizes and $\beta$.

In our sequence of target distributions we add one variable at a time and all associated factors. A simple example where we alternate left-right, right-left, can be seen in Figure 3. To be specific we choose our sequence of intermediate target distributions as

$$
\begin{aligned}
\gamma_k(X_{\mathcal{L}_k}) &\propto \gamma_{k-1}(X_{\mathcal{L}_{k-1}}) \prod_{i\in\mathcal{N}_k} e^{\beta J_{ki}\cos(x_k - x_i)} \\
&\propto \gamma_{k-1}(X_{\mathcal{L}_{k-1}}) e^{\kappa(X_{\mathcal{L}_{k-1}})\cos(x_k - \mu(X_{\mathcal{L}_{k-1}}))},
\end{aligned}
\tag{3}
$$

|     |     |     |
|-----|-----|-----|
| ① — ③ — ⑥ | ③ — ② — ① | ① — ② — ③ |
| ② — ⑤ — ⑧ | ④ — ⑨ — ⑧ | ④ — ⑤ — ⑥ |
| ④ — ⑦ — ⑨ | ⑤ — ⑥ — ⑦ | ⑦ — ⑧ — ⑨ |
| (a) Diagonal | (b) Spiral | (c) Left-Right |

Figure 1: Illustration of some of the different orderings considered in the XY model.

where $\mathcal{N}_k = \{i : (k,i) \in \mathcal{E}\} \cap \mathcal{L}_{k-1}$ denotes the set of neighbours to variable $k$ in $\mathcal{L}_{k-1}$. The quantities $\mu(X_{\mathcal{L}_{k-1}})$ and $\kappa(X_{\mathcal{L}_{k-1}})$ follow from elementary trigonometric operations (sum of cosines). From the above expression we note that, conditionally on $X_{\mathcal{L}_{k-1}}$, the variable $x_k$ is von Mises distributed under $\bar{\gamma}_k$, with $X_{\mathcal{L}_{k-1}}$-dependent mean $\mu$ and dispersion $\kappa$. This implies that we can employ full adaption of the proposed SMC sampler. This is accomplished by choosing the aforementioned von Mises distribution as proposal distribution $r_k(x_k|X_{\mathcal{L}_{k-1}})$ and by choosing the corresponding normalizing constants $\nu(X_{\mathcal{L}_{k-1}}) = 2\pi I_0(\kappa(X_{\mathcal{L}_{k-1}}))$ (where $I_0$ is the modified Bessel function of order 0) as adjustment weights. We use the fully adapted SMC sampler to estimate the partition function of the classical XY model.

We consider four different orderings of the nodes:

**Random neighbour** The first node is selected randomly among all nodes, concurrent nodes are then selected randomly from the set of nodes with a neighbour in $X_{\mathcal{L}_{k-1}}$.

**Diagonal** The nodes are added by traversing from left to right with $45°$, see Figure 1a.

**Spiral** The nodes are added spiralling in towards the middle from the edges, see Figure 1b.

**Left-Right** The nodes are added by traversing the graph from left to right, from top to bottom, see Figure 1c.

See illustrations of the node orderings displayed in Figure 1 for a $3 \times 3$ example, numbers display at what iteration the node is added.

## 1.2   Evaluation of topic models

Here we present some additional results (Figure 2) on the synthetic example for various settings of the number of topics ($T$) and words ($W$). LRS 1 and LRS 2 has 10 and 20 samples, respectively. The number of particles where set to give comparable computational complexity.

## 1.3   Gaussian Markov random field

Consider a square lattice Gaussian Markov random field (MRF) of size $10 \times 10$, given by the relation

$$p(X_{\mathcal{V}}, Y_{\mathcal{V}}) \propto \prod_{i \in \mathcal{V}} e^{\frac{1}{2\sigma_i^2}(x_i - y_i)^2} \prod_{(i,j) \in \mathcal{E}} e^{\frac{1}{2\sigma_{ij}^2}(x_i - x_j)^2}, \tag{4}$$

with latent variables $X_{\mathcal{V}} = \{x_1, \ldots, x_{100}\}$ and measurements $Y_{\mathcal{V}} = \{y_1, \ldots, y_{100}\}$. The graphical representation of the latent variables in this model is shown in Figure 3.

The measurements $Y_{\mathcal{V}}$ where simulated from the model with $\sigma_i = 1$ and $\sigma_{ij} = 0.1$. Given these measurement, we seek the posterior distribution $p(X_{\mathcal{V}} | Y_{\mathcal{V}})$. We run four different MCMC samplers to simulate from this distribution; the proposed (fully blocked) PGAS, the proposed PGAS with partial blocking, a standard one-at-a-time Gibbs sampler, and the tree-sampler proposed by Hamze and de Freitas [2004]. For the PGAS algorithms we use $N = 50$ particles. The tree-sampler exploits the fact that the model is Gaussian and it can thus not be used for arbitrary (non-Gaussian or non-discrete) graphs. By partitioning the graph into disjoint trees (in our case, chains) for which exact inference is possible, the tree-sampler implements an "ideal" partially blocked Gibbs sampler. PGAS with partial blocking can thus be seen as an SMC-based version of the tree-sampler. See Figure 3 for the ordering in the PGAS algorithm and the blocking used for tree-sampling and PGAS with partial blocking, corresponding to a partition of the graph into two chains. The variables are

(a) $T = 100, W = 20$.        (b) $T = 100, W = 50$.

(c) $T = 100, W = 100$.        (d) $T = 200, W = 10$.

Figure 2: Estimates of the log-likelihood of a synthetic LDA model.

numbered $1, \ldots, 100$ from top to bottom, left to right and $\mathcal{L}_k$ is taken as the $k$ first indices of $\mathcal{L}_K = \{1, \ldots, 10, 20, 19, \ldots, 11, 21, 22, \ldots, 100, 99, \ldots, 91\}$. This ordering gives results very similar to that of the Left-Right ordering explained above.

Figure 3: *Left:* Ordering of factors in the PGAS algorithm. At each iteration all the factors connecting the added node and the previous nodes are included in the target distribution. *Right:* The two block structures used in the tree-sampler and PGAS with partial blocking. Nodes are added from the edge spiralling in.

In Figure 4b we can see the empirical autocorrelation funtions (ACFs) centered around the true posterior mean for variable $x_{82}$ (selected randomly from $X_\mathcal{V}$). Similar results hold for all the variables of the model. Due to the strong interaction between the latent variables, the samples generated by the standard Gibbs sampler are strongly correlated. Tree-sampling and PGAS with partial blocking show nearly identical gains compared to Gibbs. This is interesting, since it suggest that simulating from the SMC-based PGAS kernel can be almost as efficient as exact simulation, even using a moderate number of particles. We emphasize that the PGAS kernels leave their respective target distributions invariant, i.e. the limiting distributions is the same for all MCMC schemes. The fully

blocked PGAS algorithm achieves the best ACF, dropping off to zero considerably faster than for the other methods. This is not surprising since this sampler simulates all the latent variables jointly which reduces the autocorrelation, in particular when the latent variables are strongly dependent.

(a) The empirical ACF for Gibbs sampler, PGAS, PGAS with partial blocking and tree sampler. Based on $100\,000$ data points with $10\%$ burnin.

(b) The empirical ACF for PGAS with partial blocking for $N = 5, 10, 20, 50$. Based on $10\,000$ data points with $10\%$ burnin.

However, this improvement in autocorrelation comes at a cost. For the fully blocked PGAS kernel, the maximal cardinality of the set $\mathcal{A}_k$ (see (9)) is 10 (one full row of variables). For the partially blocked PGAS kernel, on the other hand, $|A_k| \equiv 1$ since the variables in each block form a chain. This implies that the fully blocked PGAS sampler is an order of magnitude more computationally involved than the partially blocked PGAS sampler. This trade-off between autocorrelation and computational efficiency has to be taken into account when deciding which algorithm that is most suitable for any given problem.

## 2 Proof of unbiasedness

Recall that we use the convention $\xi_k = X_{\mathcal{I}_k \setminus \mathcal{L}_{k-1}}$. Define recursively the functions $f_k(X_{\mathcal{L}_k}) \equiv 1$ and,

$$f_\ell(X_{\mathcal{L}_\ell}) = \frac{\int f_{\ell+1}(X_{\mathcal{L}_{\ell+1}})\gamma_\ell(X_{\mathcal{L}_{\ell+1}})\mathrm{d}\xi_{\ell+1}}{\gamma_\ell(X_{\mathcal{L}_\ell})} \tag{5}$$

for $\ell = k-1, k-2, \ldots, 1$. Let

$$Q_\ell = \left(\frac{1}{N}\sum_{i=1}^{N} w_\ell^i f_\ell(X_{\mathcal{L}_\ell}^i)\right)\left\{\prod_{m=1}^{\ell-1}\frac{1}{N}\sum_{i=1}^{N}\nu_m^i w_m^i\right\},$$

for $\ell \in \{1, \ldots, k\}$. Note that, by construction, $Q_k = \widehat{Z}_k^N$. Let $\mathcal{F}_\ell$ be the filtration generated by the particles simulated up to iteration $\ell$:

$$\mathcal{F}_\ell := \sigma(\{X_{\mathcal{L}_m}^i, w_m^i\}_{i=1}^N, m = 1, \ldots, \ell).$$

**Lemma 1.** *The sequence $\{Q_\ell, \ell = 1, \ldots, k\}$ is an $\mathcal{F}_\ell$-martingale.*

*Proof.* Consider,

$$\mathbb{E}\left[Q_\ell \mid \mathcal{F}_{\ell-1}\right] = \mathbb{E}\left[w_\ell^1 f_\ell(X_{\mathcal{L}_\ell}^1) \mid \mathcal{F}_{\ell-1}\right]\left\{\prod_{m=1}^{\ell-1}\frac{1}{N}\sum_{i=1}^{N}\nu_m^i w_m^i\right\}.$$

Using the definition of the weight function (see the main document) we have,

$$
\mathbb{E}\left[W_\ell(X_{\mathcal{L}_\ell}^1)f_\ell(X_{\mathcal{L}_\ell}^1) \mid \mathcal{F}_{\ell-1}\right]
$$

$$
= \sum_{i=1}^N \int W_\ell(\{X_{\mathcal{L}_{\ell-1}}^i \cup \xi_\ell\})f_\ell(\{X_{\mathcal{L}_{\ell-1}}^i \cup \xi_\ell\})\frac{\nu_{\ell-1}^i w_{\ell-1}^i}{\sum_l \nu_{\ell-1}^l w_{\ell-1}^l}r_\ell(\xi_\ell|X_{\mathcal{L}_{\ell-1}}^i)\mathrm{d}\xi_\ell
$$

$$
= \frac{1}{\sum_l \nu_{\ell-1}^l w_{\ell-1}^l}\sum_{i=1}^N \left(\nu_{\ell-1}^i w_{\ell-1}^i \frac{\int \gamma_\ell(\{X_{\mathcal{L}_{\ell-1}}^i \cup \xi_\ell\})f_\ell(\{X_{\mathcal{L}_{\ell-1}}^i \cup \xi_\ell\})\mathrm{d}\xi_\ell}{\gamma_{\ell-1}(X_{\mathcal{L}_{\ell-1}}^i)\nu_{\ell-1}(X_{\mathcal{L}_{\ell-1}}^i)}\right)
$$

$$
= \frac{1}{\sum_l \nu_{\ell-1}^l w_{\ell-1}^l}\sum_{i=1}^N \left(w_{\ell-1}^i f_{\ell-1}(X_{\mathcal{L}_{\ell-1}}^i)\right).
$$

Hence, we get

$$
\mathbb{E}\left[Q_\ell \mid \mathcal{F}_{\ell-1}\right] = \sum_i \left(w_{\ell-1}^i f_{\ell-1}(X_{\mathcal{L}_{\ell-1}}^i)\right)\frac{1}{N}\left\{\prod_{m=1}^{\ell-2}\frac{1}{N}\sum_i \nu_m^i w_m^i\right\} = Q_{\ell-1}.
$$

$\square$

It follows that

$$
\mathbb{E}[\widehat{Z}_k^N] = \mathbb{E}[Q_k] = \mathbb{E}[Q_1] = \int W_1(X_{\mathcal{L}_1})f_1(X_{\mathcal{L}_1})r_1(X_{\mathcal{L}_1})\mathrm{d}X_{\mathcal{L}_1} = \int \gamma_1(X_{\mathcal{L}_1})f_1(X_{\mathcal{L}_1})\mathrm{d}X_{\mathcal{L}_1}.
$$

However, from the definition in (5) we have that

$$
\int \gamma_1(X_{\mathcal{L}_1})f_1(X_{\mathcal{L}_1})\mathrm{d}X_{\mathcal{L}_1} = \iint \gamma_2(X_{\mathcal{L}_2})f_2(X_{\mathcal{L}_2})\mathrm{d}X_{\mathcal{L}_2}
$$

$$
= \cdots = \int\cdots\int \gamma_k(X_{\mathcal{L}_k})f_k(X_{\mathcal{L}_k})\mathrm{d}X_{\mathcal{L}_k} = Z_k.
$$

$\square$

## 3  Ancestor sampling

To implement the PGAS sampling procedure, it remains to detail the ancestor sampling step. At each iteration $k \geq 2$, this step amount to generating a value for the ancestor index $a_k^N$ corresponding to the reference particle. Implicitly, this assigns an artificial history for the "remaining" part of the reference particle $X'_{\mathcal{L}_K \setminus \mathcal{L}_{k-1}}$, by selecting one of the particles $\{X_{\mathcal{L}_{k-1}}^i\}_{i=1}^N$ as its ancestor. This results in a complete assignment for the collection of latent variables of the PGM

$$
\widetilde{X}_{\mathcal{L}_K}^i := \{X_{\mathcal{L}_{k-1}}^i \cup X'_{\mathcal{L}_K \setminus \mathcal{L}_{k-1}}\} \in \mathsf{X}_\mathcal{V}. \tag{6}
$$

As shown by Lindsten et al. [2014], the probability distribution from which $a_k^N$ should be sampled in order to ensure reversibility of the PGAS kernel w.r.t. $\bar{\gamma}_K$ is given by,

$$
\mathbb{P}(a_k^N = i) \propto w_{k-1}^i \frac{\gamma_K\left(\widetilde{X}_{\mathcal{L}_K}^i\right)}{\gamma_{k-1}(X_{\mathcal{L}_{k-1}}^i)}. \tag{7}
$$

This expression can be understood as an application of Bayes' theorem, where $w_{k-1}^i$ is the prior probability of particle $X_{\mathcal{L}_{k-1}}^i$ and the ratio between the target densities is the unnormalized likelihood of $X'_{\mathcal{L}_K \setminus \mathcal{L}_{k-1}}$ conditionally on $X_{\mathcal{L}_{k-1}}^i$.

To derive an explicit expression for the ancestor sampling probabilities in our setting, note first that,

$$
\frac{\gamma_K\left(X_{\mathcal{L}_K}\right)}{\gamma_{k-1}\left(X_{\mathcal{L}_{k-1}}\right)} = \frac{\prod_{j=k}^K \psi_j\left(X_{\mathcal{I}_j}\right)}{q_{k-1}(X_{\mathcal{L}_{k-1}})}. \tag{8}
$$

Now, let $\mathcal{A}_k$ be the index set of factors $\psi_j$, $j \geq k$ for which any of the variables $X_{\mathcal{L}_{k-1}}$ is in the domain of $\psi_j$; formally

$$\mathcal{A}_k := \{j : k \leq j \leq K, \mathcal{L}_{k-1} \cap \mathcal{I}_j \neq \emptyset\}. \tag{9}$$

It follows that any factor $\psi_j$ for which $j \notin \mathcal{A}_k$ is independent of $X_{\mathcal{L}_{k-1}}$. Consequently, we can write (7) as

$$\mathbb{P}(a_k^N = i) \propto w_{k-1}^i \frac{\prod_{j \in \mathcal{A}_k} \psi_j \left( \widetilde{X}_{\mathcal{I}_j}^i \right)}{q_{k-1}(X_{\mathcal{L}_{k-1}}^i)}, \tag{10}$$

where $\widetilde{X}_{\mathcal{I}_j}^i$ is a subset of the variables in (6). In fact, the index set $\mathcal{A}_k$ corresponds exactly to the factors $\psi_j$ that depend, explicitly, both on the particle $X_{\mathcal{L}_{k-1}}^i$ and on the reference particle $X'_{\mathcal{L}_K \setminus \mathcal{L}_{k-1}}$ (through some of their respective components). Indeed, it is only these factors that hold any information about the likelihood of $X'_{\mathcal{L}_K \setminus \mathcal{L}_{k-1}}$ given $X_{\mathcal{L}_{k-1}}^i$.

The expression (10) is interesting, since it shows that the computational complexity of the ancestor sampling step will depend on the cardinality of the set $\mathcal{A}_k$. While this will depend both on the structure of the graph and on the ordering of the factors, it will for many models of interest be of a lower order than the cardinality of $\mathcal{V}$.