[Reviews · NeurIPS 2014]

Submitted by Assigned_Reviewer_34

This paper is concerned with Sequential Monte Carlo Methods for Probabilistic Graphical Models (PGM). The main contribution of this paper is that it introduces a sequence of auxiliary distributions defined on a monotonically increasing sequence of probability spaces. The authors make use of the structure of the PGM to define a sequence of intermediate target distributions for the sampler. The SMC sampler that is proposed can be then used within a Particle MCMC algorithm to come with efficient algorithms both for parameter and state estimation.

The paper is generally well written and the authors do a good job to explain what the algorithm is and illustrate its performance versus existing algorithms.
Summary: This is a nicely written paper which proposes a novel efficient SMC sampler for Probabilistic Graphical Models and its performance is illustrated via various examples.

Submitted by Assigned_Reviewer_42

The authors propose a method for applying sequential Monte Carlo (SMC) inference to probabilistic graphical models (PGMs). Specifically, the authors show how SMC can be applied to infer latent variables and estimate the partition function (model evidence, posterior normalisation) when a PGM is represented as a factor graph. The authors also show how SMC inference for PGMs can be used as a kernel in an MCMC sampler (via PMCMC) to perform block sampling of latent variables. Finally, the authors apply their method to several datasets showing that their method can i) accurately estimate the partition function and ii) PMCMC sampling can be used to introduce block resampling moves of latent variables with very low auto-correlation.

Using SMC for inference in PGMs is an interesting idea that exploits the fact that SMC inference can be done on fairly arbitrary sequence of distributions, provided that the final distribution in the sequence is the desired target distribution. The authors clearly explain how the factor graph representation of a PGM can be decomposed to define such a sequence of distributions.

I found the review of factor graphs well written but lacking in references to more detailed resources. I felt a similar section reviewing SMC inference, independent of the application to PGMs would greatly aid paper. As the paper is written it may be difficult for readers without a fairly detailed knowledge of the SMC literature to access the paper. For example the authors talk about adjustment multipliers and fully adapted samplers in section 3.2. While these are important concepts for building efficient SMC samplers, they are also potentially unnecessary complications when presenting the basic idea of SMC inference in PGMs.

The experiment in section 5.1 is sound and demonstrates that the estimates of the partition function generated by SMC are comparable to state of the art methods.

I am not satisfied with the results of section 5.2. I found it surprising that the authors were able to include $\theta$ as part of the SMC chain, particularly that the variable would be added first during inference. For the synthetic dataset (Figure 6a) the authors show that the SMC algorithm performs reasonably well. I suspect this is due to the small number of topics and the corresponding small dimension of $\theta$. It would be useful to repeat the experiment with a larger number of topics so that the dimension of $\theta$ is larger and see if performance remains competitive with LRS. It is also not clear how the authors are determining the performance on the real data (Figures 6b-c); are they comparing against LRS or looking at the variability of estimates?

The experiment in section 5.3 explores the use of PMCMC for PGMs. Building PMCMC samplers that can update blocks of latent variables by SMC and parameters by Metropolis-Hastings or Gibss sampling could be very powerful. Unfortunately, I feel the experiment is incomplete. The authors only examine auto-correlation times (Figure 7), but do not show a comparison of time complexity of the iterations for each method, or how accurate the different inference schemes are for a fixed computational budget.

A few other small issues

- For figures 3 and 6a I am not sure what N on the x-axis refers to.

- The sentence "Both methods converges to the true value, however, despite the fact that it uses much fewer computational SMC performs significantly better in terms of estimator variance." on lines 369-371 is grammatically incorrect and confusing.

- In supplemental section 1.1 I couldn't find the definition for the functions $\kappa$ or $\mu$. They are defined implicitly as the mean and dispersion of a Von Mise distribution, but it would be useful to explicitly state what they are.
Summary: This work presents a novel and interesting inference scheme for PGMs. However, the current presentation of the model is not sufficiently clear and the benchmarking is not adequate.

Submitted by Assigned_Reviewer_43

The authors apply a Sequential Monte Carlo sampling algorithm in the context of graphical models. The key idea is to construct an artificial sequence of auxiliary distributions that build up to the target distribution of interest; the authors do this through a factor graph representation of the target graphical model. Whilst the authors refer to an intuitive ordering of the sequence of artificial targets used in the general algorithmic case, I would have found it more instructive to include a detailed discussion on this point within Section 3. This appears to be a key issue and feels like it is superficially treated in the paper as it stands. As a natural extension, the authors offer a Particle MCMC-based algorithm that exploits their SMC sampler based technique. Section 4 also requires additional detail on the partial blocked decomposition definition. The reader is left with vague statements, which are only partially illuminated during the experiments section. The experiments section is well written and considers a good range of example models. I believe that this is a very good contribution to the scientific literature. The paper is well written, the references are complete, and the presentation is accurate.
Summary: I believe that this paper should be published at NIPS. In my opinion it will be of interest to the scientific community.
Author Feedback
Author rebuttal: We would like to thank the reviewers for the many insightful comments and valuable suggestions for improvements. We will make sure to take these into account.

To clarify the LDA likelihood estimation example (Section 5.2), we use marginalisation of $\theta$ within the SMC algorithm (cf. the L-R [15], and LRS [16] methods) and a fully adapted SMC sampler. This mitigates the degeneracy issue, although we are aware that it is not completely resolved. Nevertheless, the intention with this example was to show that even an out-of-the-box application of SMC can give comparable results to a special-purpose state-of-the-art method, such as LRS.

We agree that repeating the simulation experiment with a larger number of topics is of interest.
The comment raised in the review report motivated us to run additional simulations with a larger number of topics (up to 200) and words (up to 100). These simulations resulted in similar performance gains for SMC compared to LRS as those reported for the smaller example in the paper.

Regarding the real data LDA analysis (also Section 5.2): Unfortunately (and embarrassingly), we realized after submission that we had included the wrong plot in Fig. 6(c). We sincerely apologize for the confusion caused by this mistake, as expressed by Assigned_Reviewer_42. The correct values for the estimates of \log(Z) for Fig. 6(c) are:

(mean, std from bootstrapping)
LRS 1: (-13540, 11)
LRS 2: (-13508, 9)
SMC 1: (-13509, 10)
SMC 2: (-13496, 11)

The performance of the algorithms on the real data is assessed by comparing the mean values of the estimates. By computing the logarithm of the normalizing constant estimate, a negative bias is introduced (cf. Jensen’s inequality and recall that SMC gives an unbiased estimate of Z). Consequently, larger values of \log(Z) typically implies more accurate results. Based on this criteria and the numbers reported above, we see that SMC gives slightly better results than LRS also for the ‘20 newsgroups’ data (SMC1 and LRS2 give similar performance, and SMC2 gives the overall most accurate results).

The Gaussian MRF example (Section 5.3) was included primarily as a proof-of-concept for PMCMC and not to reflect any actual application where the use of PMCMC is justified. In particular, we wanted to illustrate the potential of mimicking the tree-sampler [30] using PMCMC, thereby extending the scope of this powerful method to PGMs where it is otherwise not applicable.